

# Identification of risk factors for mortality associated with COVID-19

Yuetian Yu[1,*], Cheng Zhu[2,*], Luyu Yang[3], Hui Dong[3], Ruilan Wang[4], Hongying Ni[5], Erzhen Chen[2] and Zhongheng Zhang[6]

[1] Department of Critical Care Medicine, Ren Ji Hospital, School of Medicine, Shanghai Jiao Tong University, Shanghai, China
[2] Department of Emergency Medicine, Rui Jin Hospital, School of Medicine, Shanghai Jiao Tong University, Shanghai, China
[3] Department of Intensive Care Unit, Wuhan Third Hospital, Wuhan University, Wuhan, China
[4] Department of Critical Care Medicine, Shanghai General Hospital, Shanghai Jiao Tong University, Shanghai, China
[5] Department of Critical Care Medicine, Jinhua Municipal Central Hospital, Jinhua, Zhejiang, China
[6] Department of Emergency Medicine, Sir Run Run Shaw hospital; Zhejiang University School of Medicine, Hangzhou, China
* These authors contributed equally to this work.

Corresponding authors
Erzhen Chen,
erzhenchen1963@sina.com
Zhongheng Zhang,
zh_zhang1984@zju.edu.cn

## ABSTRACT

**Objectives:** Coronavirus Disease 2019 (COVID-19) has become a pandemic outbreak. Risk stratification at hospital admission is of vital importance for medical decision making and resource allocation. There is no sophisticated tool for this purpose. This study aimed to develop neural network models with predictors selected by genetic algorithms (GA).

**Methods:** This study was conducted in Wuhan Third Hospital from January 2020 to March 2020. Predictors were collected on day 1 of hospital admission. The primary outcome was the vital status at hospital discharge. Predictors were selected by using GA, and neural network models were built with the cross-validation method. The final neural network models were compared with conventional logistic regression models.

**Results:** A total of 246 patients with COVID-19 were included for analysis. The mortality rate was 17.1% (42/246). Non-survivors were significantly older (median (IQR): 69 (57, 77) vs. 55 (41, 63) years; $p < 0.001$), had higher high-sensitive troponin I (0.03 (0, 0.06) vs. 0 (0, 0.01) ng/L; $p < 0.001$), $C$-reactive protein (85.75 (57.39, 164.65) vs. 23.49 (10.1, 53.59) mg/L; $p < 0.001$), $D$-dimer (0.99 (0.44, 2.96) vs. 0.52 (0.26, 0.96) mg/L; $p < 0.001$), and $\alpha$-hydroxybutyrate dehydrogenase (306.5 (268.75, 377.25) vs. 194.5 (160.75, 247.5); $p < 0.001$) and a lower level of lymphocyte count (0.74 (0.41, 0.96) vs. 0.98 (0.77, 1.26) $\times 10^9$/L; $p < 0.001$) than survivors. The GA identified a 9-variable (NNet1) and a 32-variable model (NNet2). The NNet1 model was parsimonious with a cost on accuracy; the NNet2 model had the maximum accuracy. NNet1 (AUC: 0.806; 95% CI [0.693–0.919]) and NNet2 (AUC: 0.922; 95% CI [0.859–0.985]) outperformed the linear regression models.

**Conclusions:** Our study included a cohort of COVID-19 patients. Several risk factors were identified considering both clinical and statistical significance. We further developed two neural network models, with the variables selected by using GA. The model performs much better than the conventional generalized linear models.

## INTRODUCTION

The Coronavirus Disease 2019 (COVID-19) pandemic outbreak has become a global health emergency since its outbreak in Wuhan, China, and it is now spreading rapidly across the world (*Huang et al., 2020*; *Ren et al., 2020*). More recently, the World Health Organization declared it to be a pandemic outbreak due to its capability of human-to-human transmission and rapid spread over the globe. The COVID-19-specific mortality rate has been reported to be from 2% to 20% (*Sun, Chen & Viboud, 2020*; *Yang et al., 2020*; *Chen et al., 2020b*), depending on the availability of medical resources and economic status. One of the most important issues in managing COVID-19 is the accurate and early identification of high-risk patients. Early risk stratification can help medical decision making and resource allocation, for example, high-risk patients can be transferred to the intensive care unit for close monitoring and organ support. Although several studies have investigated the risk factors for mortality in COVID-19 (*Liu et al., 2020*; *Wang et al., 2020*), there has been no systematic effort to develop a prediction tool for risk stratification at an early stage.

Conventionally, prediction models are developed with generalized linear models, which, however, cannot capture the non-linear association between covariates (*Friedman, 2010*; *Tolles & Meurer, 2016*). In the era of big data, a large volume of data can be obtained from electronic healthcare records (EHR), which causes the curse of dimensionality. In this study, we extracted variables from the EHR and developed a neural network model with covariates selected by genetic algorithms (GA) (*Tolvi, 2004*). The benefit of using this approach is that it automatically captures the non-linear and interaction terms. We also showed that the neural network models performed better than conventional generalized linear models.

## METHODS

### Study design and setting

The study was conducted in Wuhan Third Hospital from January 2020 to February 2020. All COVID-19 patients treated in our hospital during the study period were considered for inclusion. No further new cases were enrolled after February 2020. The EHRs of subjects with confirmed COVID-19 were reviewed retrospectively. Patients were divided into the survival and non-survival groups depending on vital status at hospital discharge. Patients were followed for 30 days for their vital status if they discharged earlier than 30 days. All-cause mortality was considered as the study end point. All laboratory tests and baseline medical history were extracted on day 1 of hospital admission. Neural network models were developed to predict in-hospital mortality. The study has been approved by the ethics committee of Wuhan Third Hospital (KY2020-007). Informed consent was waived as determined by the institutional review board due to the retrospective study design. The study was conducted in accordance to the Helsinki Declaration. The study was

reported as per the STrengthening the Reporting of OBservational studies in Epidemiology (STROBE) checklist (*Von Elm et al., 2007*).

## Participants

All patients confirmed to have COVID-19 were included for analysis. A patient was suspected to have COVID-19 if he/she satisfied the diagnostic criteria by clinical features and epidemiological risks. Clinical criteria included fever and signs/symptoms (e.g., cough or shortness of breath) consistent with a lower respiratory illness. Epidemiologic risks were (1) within 14 days of symptom onset, there is evidence of close contact with a laboratory-confirmed 2019-nCoV patient; (2) a traveling history to Hubei Province within 14 days of symptom onset. A patient with above mentioned evidence could be further confirmed to have COVID-19 if one of the following criteria was satisfied: (1) positive nucleic acid test for novel coronavirus by real-time (RT)-PCR in blood or respiratory specimens; and (2) a homogenous sequence consistent with the novel coronavirus as identified by genetic sequencing (*Jin et al., 2020*). The inclusion criteria were described before (*Hong et al., 2020*). Patients who had missing values on >80% of variables, had severe trauma injury, signed a do-not-resuscitate order or were younger than 18 years old were excluded from analysis.

## Variables

Demographic data such as sex, age, weight and height were collected. Comorbidities were classified by system such as respiratory system, digestive system, endocrinology and metabolism, muscle and skeleton, and reproductive system. Medical histories of surgery and infectious disease were extracted. Vital signs such as respiratory rate, temperature, pulse rate and blood pressure were recorded on admission. Laboratory tests such as routine blood count, chemistry profile, coagulation profile, electrolytes and brain natriuretic peptide (BNP) were included. All variables were extracted during the first 24 h following hospital admission. If there were multiple measurements of a variable, the earliest one was used for analysis. Variables with more than 30% missing values across patients were excluded. The remaining variables with missing values were imputed with a random sample of the remaining complete values (*Zhang, 2016a*).

## Statistical analysis

Descriptive statistics were performed by conventional methods. Briefly, mean and standard distribution were used for the expression of normally distributed data and the survival and non-survival groups were compared for these variables with $t$ tests. Median and interquartile range (IQR) were used for non-normal data and between-group comparisons were performed with the rank-sum test. Categorical variables were compared between groups using Chi-square or Fisher's exact test as appropriate. The analyses were performed as described before (*Zhang et al., 2017*; *Hong et al., 2020*).

A genetic algorithm is a type of evolutionary computer algorithm in which symbols (i.e., "genes" or "chromosomes") representing possible solutions are "bred". This "breeding" of symbols includes the utilization of crossing-over process in genetic

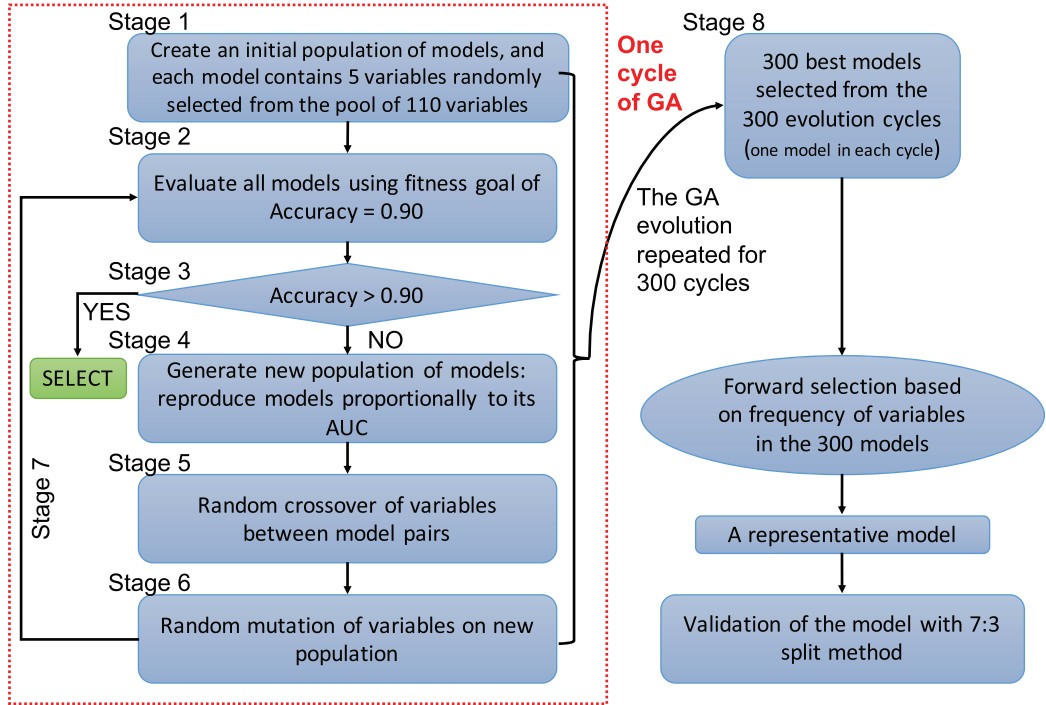

**Figure 1 Flowchart of genetic algorithms.** One cycle of the GA represents an evolution cycle that comprised seven stages, as shown in the figure. Each cycle developed the best model. Our study ran 300 evolutionary cycles, resulting in the 300 best fitting chromosomes (neural network models). Each model contained five predictors. Forward selection was performed to develop a representative model(s). Finally, the representative model(s) were validated and compared with conventional logistic regression models.

recombination and an adjustable mutation rate. A fitness function is then used to gradually improve the solutions on each generation of algorithms in analogy to the process of natural selection (*Tolvi, 2004*; *Trevino & Falciani, 2006*). The process of evolving the GA and automating the selection is known as genetic programing. In the present study, each clinical variable was a gene and formed a chromosome (i.e., a combination of variables) (Fig. 1). The fitness function is the accuracy, which is calculated by the correctly predicted samples divided by the total samples. Details of GA can be found in the supplemental digital content (SDC).

Variables selected by the GA were used to develop a neural network model (*Patel & Goyal, 2007*; *Zhang, 2016b*). A neural network can be thought of as a network of "neurons" that are organized in layers. The predictors (or inputs) form the bottom layer, and the forecasts (or outputs) form the top layer. There may also be intermediate layers containing "hidden neurons". Details of hyperparameter tuning and construction can be found in the SDC (Fig. S7). Five-fold cross-validation was employed to prevent overfitting of the model. The variables selected by the GA were also used to construct a conventional logistic regression model, and the predictive performance was compared with the neural network models. The comparison was performed by using 70% samples as the training set and the remaining 30% samples as the validation set. That is, four models were trained on the training set, and they were validated on the remaining 30% samples to calculate

the area under the receiver operating characteristic curve (AUC). Delong's method was used to compare the AUCs (*DeLong, DeLong & Clarke-Pearson, 1988*).

Finally, the neural network models were interpreted by using local interpretable model-agnostic explanations (LIME). Essentially, LIME interprets complex models by providing a qualitative link between the predictors and the outcome. The LIME algorithm is accomplished by locally approximating the more complex model with simpler models, such as generalized linear models. The simpler models are conceptually easier to understand for subject-matter audience (*Ribeiro, Singh & Guestrin, 2016*; *Zhang et al., 2018*).

# RESULTS

## Participants and descriptive data

We initially identified 276 patients who had a confirmed diagnosis of COVID-19. After excluding 30 patients with missing values, we ultimately obtained 246 subjects for analysis.

The overall mortality rate was 17.1% (42/246) in the study cohort. Non-survivors were significantly older (median (IQR): 69 (57, 77) vs. 55 (41, 63) years; $p < 0.001$), had higher high-sensitivity troponin I (0.03 (0, 0.06) vs. 0 (0, 0.01) ng/l; $p < 0.001$), $C$-reactive protein (85.75 (57.39, 164.65) vs. 23.49 (10.1, 53.59) mg/l; $p < 0.001$), $D$-dimer (0.99 (0.44, 2.96) vs. 0.52 (0.26, 0.96) mg/L; $p < 0.001$), and $\alpha$-hydroxybutyrate dehydrogenase (306.5 (268.75, 377.25) vs. 194.5 (160.75, 247.5) mmol/L; $p < 0.001$) and a lower lymphocyte count (0.74 (0.41, 0.96) vs. 0.98 (0.77, 1.26) $\times 10^9$/L; $p < 0.001$) than hospital survivors (SDC Table S1; Fig. S1). As expected, the non-survival group had more comorbidities in the respiratory system (12% vs. 4%; $p = 0.069$) and circulatory system (50% vs. 21%; $p < 0.001$), and they had a past history of infectious disease (12% vs. 2%; $p = 0.009$) and trauma (10% vs. 2%; $p = 0.031$). Volcano plots can help to visualize both the statistical and clinical significance of a biomarker between died and alive groups (Fig. 2).

## Variable selection with GA

The variable selection process is shown in Fig. 3. A total of 300 evolution processes were performed. Each evolution cycle developed the chromosome (a combination of variables) that had the best predictive performance. A maximum of 200 generations were allowed in each evolution cycle. If the fitness goal (>90% accuracy) was reached before 200 generations, the evolution cycle was terminated, and the program went into the next cycle of evolution. Figure 2 shows the most frequently appearing variables in selected chromosomes. The top six variables selected as ranked by their frequency of appearance were age, BNP, $D$-dimer, platelet count, average platelet volume and urea nitrogen (Fig. 3).

The above GA procedure provided a large collection of chromosomes with good predictive accuracy. We need to develop a representative model for the classification problem. A forward selection procedure was employed for this purpose (Fig. S2). The procedure resulted in 19 models whose predictive accuracy was higher than 99% of the maximum. Model 15 was the model with the best predictive accuracy. However, the first
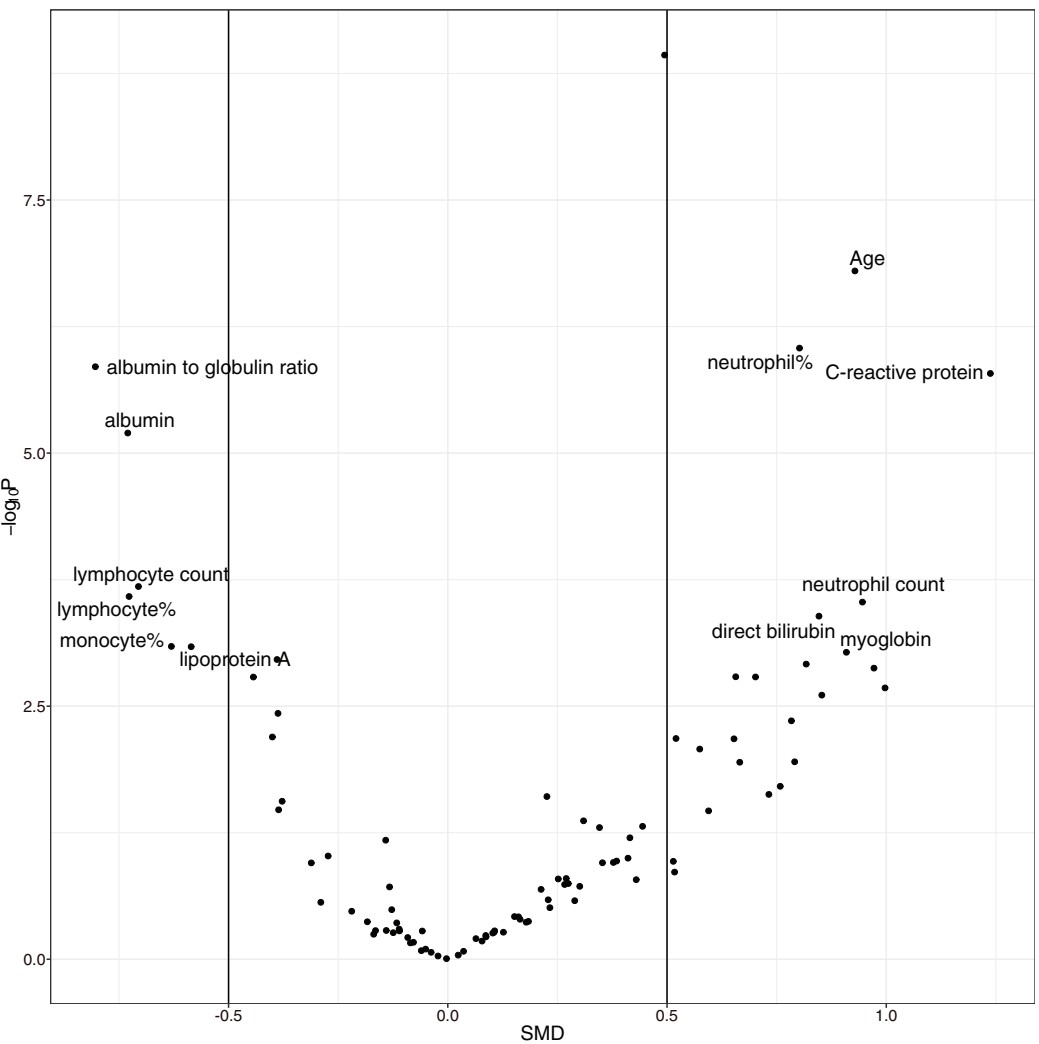

**Figure 2 Volcano plot showing significantly different variables between survivors and non-survivors.** The vertical axis is the statistical significance, and the horizontal axis represents the clinical significance. SMD, standardized mean difference.

model with only nine variables was also chosen based on the principle of parsimony. As a result, two models were generated in this step: NNet1 contains nine variables, including age, BNP, urea nitrogen, total platelet count, average platelet volume, *D*-dimer, high-sensitivity troponin I, LDH and creatinine kinase isoenzyme. The second model NNet2 contains 32 variables (Fig. S2). The two models can be visualized with heatmaps, and their ability to classify live vs. dead patents can be visualized with a principal component analysis (PCA) plot (SDC Figs. S3–S6).

## Neural network model training

To evaluate the performance of NNet1 and NNet2, they were trained and validated with 5-fold cross-validation. The hyperparameter tuning is shown in SDC Fig. S7. The neural networks were fully connected perceptrons with one hidden layer. The number of hidden units were four and 10 in the NNet1 and NNet2, respectively. The hyperparameters
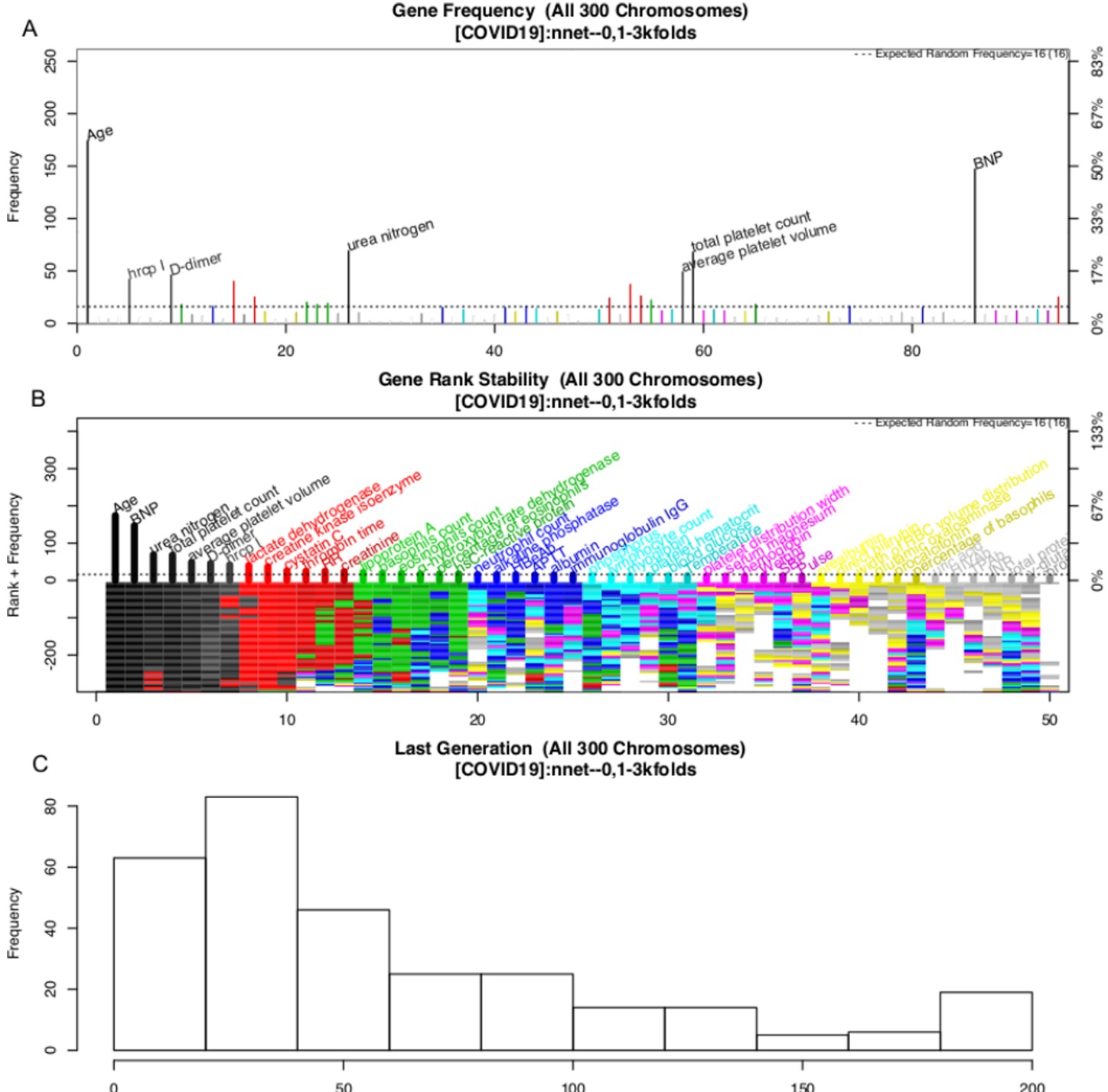

**Figure 3 Monitoring the 300 evolution cycles with each cycle producing one chromosome.** (A) The frequency by which each gene has been present in the best chromosome obtained in each evolution cycle. The top 50 genes are denoted with colors, and the top seven are annotated with names. (B) The changes of the rank of the top 50 genes with evolution cycles. The plot shows different colors when genes have many changes in ranks, indicating the rank of these genes is unstable. In the example, the top seven "black" genes are stabilized quickly in 100 evolutions, whereas "grey" genes showed unstable ranks. (C) The histogram of the number of generations required to reach the fitness goal across evolutions.

**Table 1 Parsimonious logistic regression model for the prediction of mortality.**

| Variables | OR [95% CI] | p value |
|---|---|---|
| Age | 1.09 [1.04–1.14] | <0.001 |
| BNP | 1.00 [0.99–1.00] | 0.823 |
| Urea nitrogen | 1.08 [0.96–1.24] | 0.215 |
| Total platelet count. | 1.00 [0.99–1.01] | 0.840 |
| Average platelet volume. | 1.45 [0.91–2.67] | 0.208 |
| D-dimer | 1.07 [1.00–1.16] | 0.071 |
| Hypersensitive troponin I | 1.12 [1.01–4.74] | 0.009 |
| Lactate dehydrogenase | 1.01 [1.00–1.01] | 0.003 |
| Creatine kinase isoenzyme | 1.13 [1.06–1.19] | 0.023 |

**Note:**
BNP, B-type natriuretic peptide; CI, confidence interval; OR, Odds ratio.

with the best predictive accuracy were used to develop the neural network models. Two logistic regression models (Table 1), Logit1 and Logit2, were developed by using variables included in NNet1 and NNet2, respectively. The diagnostic performance of these models was evaluated by the AUCs (Fig. 4A). The NNet1 model (AUC: 0.806; 95% CI [0.693–0.919]) and NNet2 (AUC: 0.922; 95% CI [0.859–0.985]) outperformed the Logit1 (AUC: 0.744; 95% CI [0.577–0.911]) and Logit2 (AUC: 0.802; 95% CI [0.631–0.973]). The DeLong's test showed that the NNet2 model outperformed Logit2 with statistical significance ($p = 0.011$). NNet1 was significantly better than Logit1 ($p = 0.021$). But NNet1 was not significantly better than Logit2 ($p = 0.324$). Variable importance can be calculated by identifying all connections between each predictor and the outcome. Pooling and scaling all weights specific to a predictor generates a single value ranging from 0 to 100 that reflects relative predictor importance. The variable importance makes the predictors comparable to each other (*Garson, 1991*). Figures 4B and 4C show the variable importance in NNet1 and NNet2.

## Model interpretation with LIME

Although neural network models are superior to generalized linear models in prediction accuracy, they suffer from a limitation: the black-box property means that their interpretation is not straightforward. LIME was used to interpret the NNet1 model (Fig. 5). The figure illustrates how to illustrate the NNet1 model by subject-matter audience. Four subjects including two survivors and two non-survivors are illustrated in the figure. The supporting and contradicting features used to make a mortality prediction are shown in the figure. Case 1 was alive at hospital discharge. The features of total platelet count >237 ×$10^9$/L, LDH < 350 mmol/L and BNP < 46.45 ng/l support the survival outcome. The features with red bars contradict the outcome. More categorized features used in the prediction of an alive vs. dead outcome are shown in Fig. S8.

## Comparison with other machine learning models

This section compared the predictive performance of different machine learning methods, namely, the AdaBoosting model, support vector machine (SVM), neural network

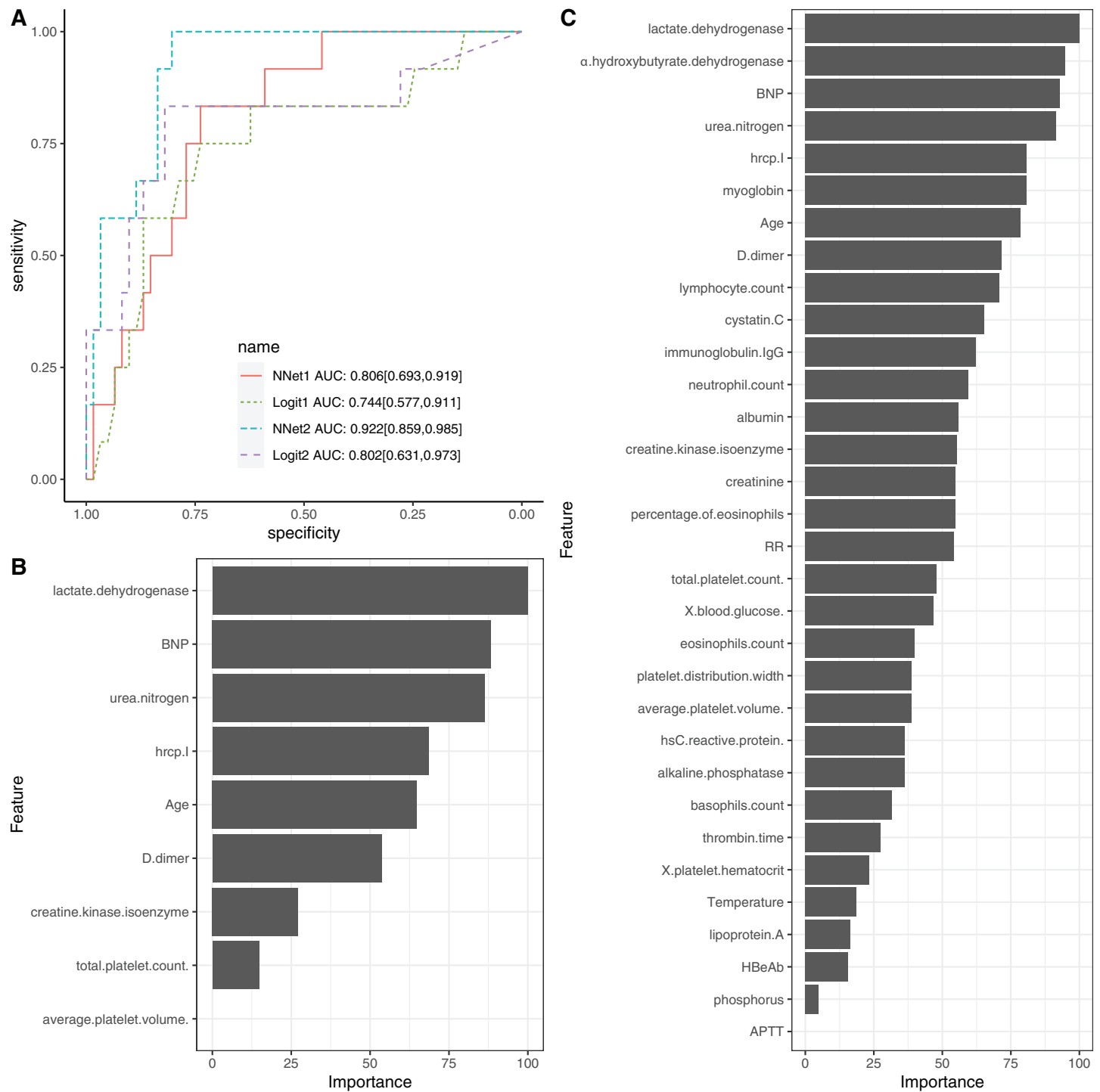

**Figure 4 Neural network models.** (A) The comparison of discriminations with AUC between four models. NNet1 is the neural network model including nine predictors, and it outperforms the logistic regression models (Logit1 and Logit2). NNet2 is the neural network model including 32 predictors, and it outperforms the other three models. The variable importance for NNet1 (B) and NNet2 (C) is shown.

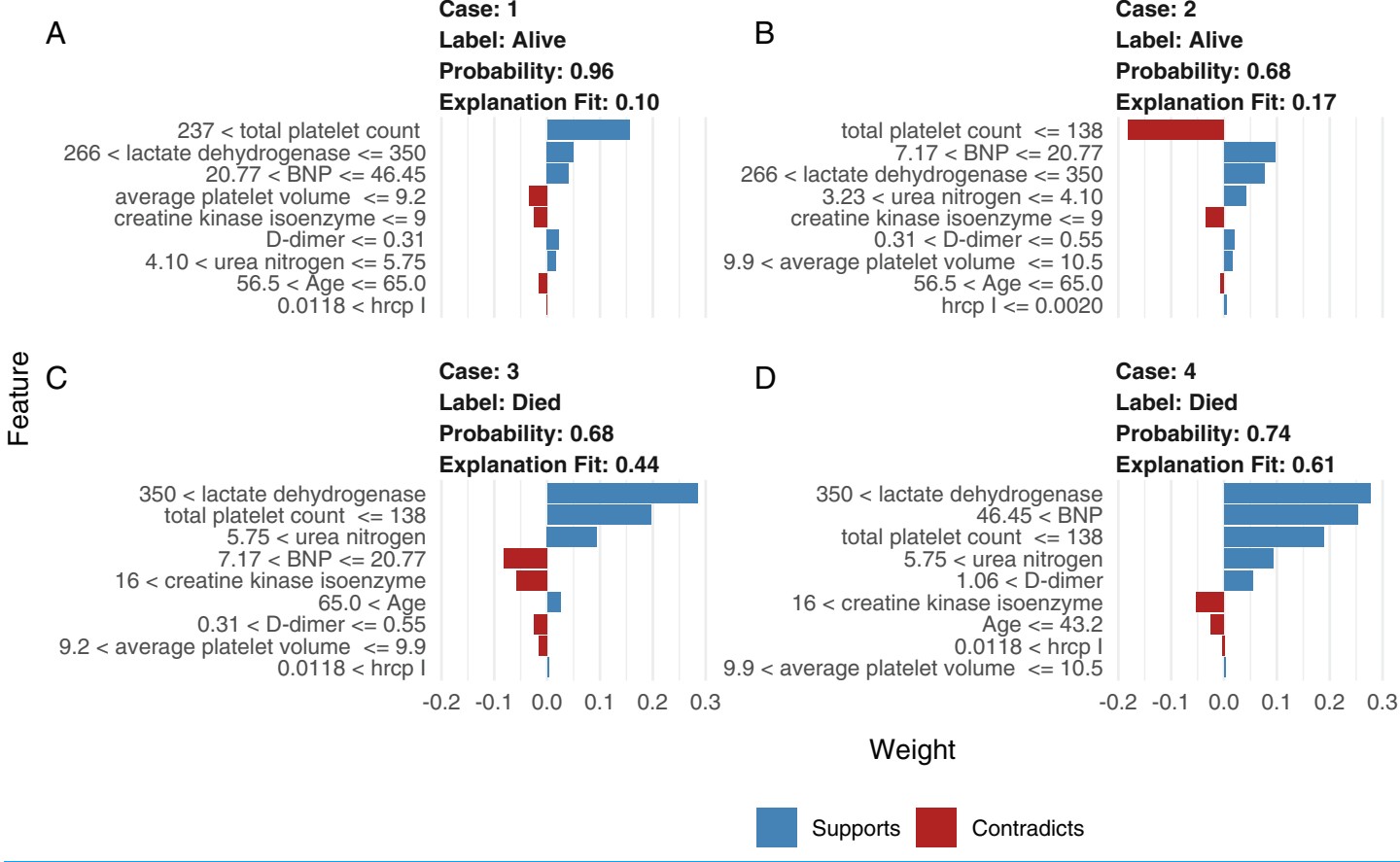

**Figure 5 Interpretation of the neural network (NNet2) with local interpretable model-agnostic explanations.** Two alive (A and B) and two dead (C and D) patients are illustrated. Case 1 (A) was alive at hospital discharge. The features of total platelet count >237 × 10⁹/L, LDH < 350 mmol/L, and BNP < 46.45 ng/l support the survival outcome. The features with red bars contradict the outcome.

model and logistic regression model (Fig. 6). In this section, all variables were entered the models, instead of using genetic algorithm for variable filtering. The result showed that the predictive performance of AdaBoosting and NNet model were not significantly different (AUC: 0.869 vs. 0.891, $p = 0.091$ for Delong's test). NNet was better than the SVM (AUC: 0.891 vs. 0.825; $p = 0.012$) and logistic regression model (AUC: 0.891 vs. 0.743; $p = 0.011$; Fig. 7). The reason for the neural network to outperform SVM in our case is probably due to the fact that the NNet model is fixed in terms of its inputs nodes, hidden layers, and output nodes; in a SVM, however, the number of support vector lines could reach the number of instances in the worst case. In this case, the SVM may overfit the data because the sample size in our study is relatively small. The NNet model performs better than the logistic regression because the former is able to automatically handle high-order interaction and non-linear terms (*Dreiseitl & Ohno-Machado, 2002*). Since we included many predictors for model training, it is very probably that some of these variables are related to each other with complex functional forms. The logistic regression model cannot capture these forms by model specification manually.

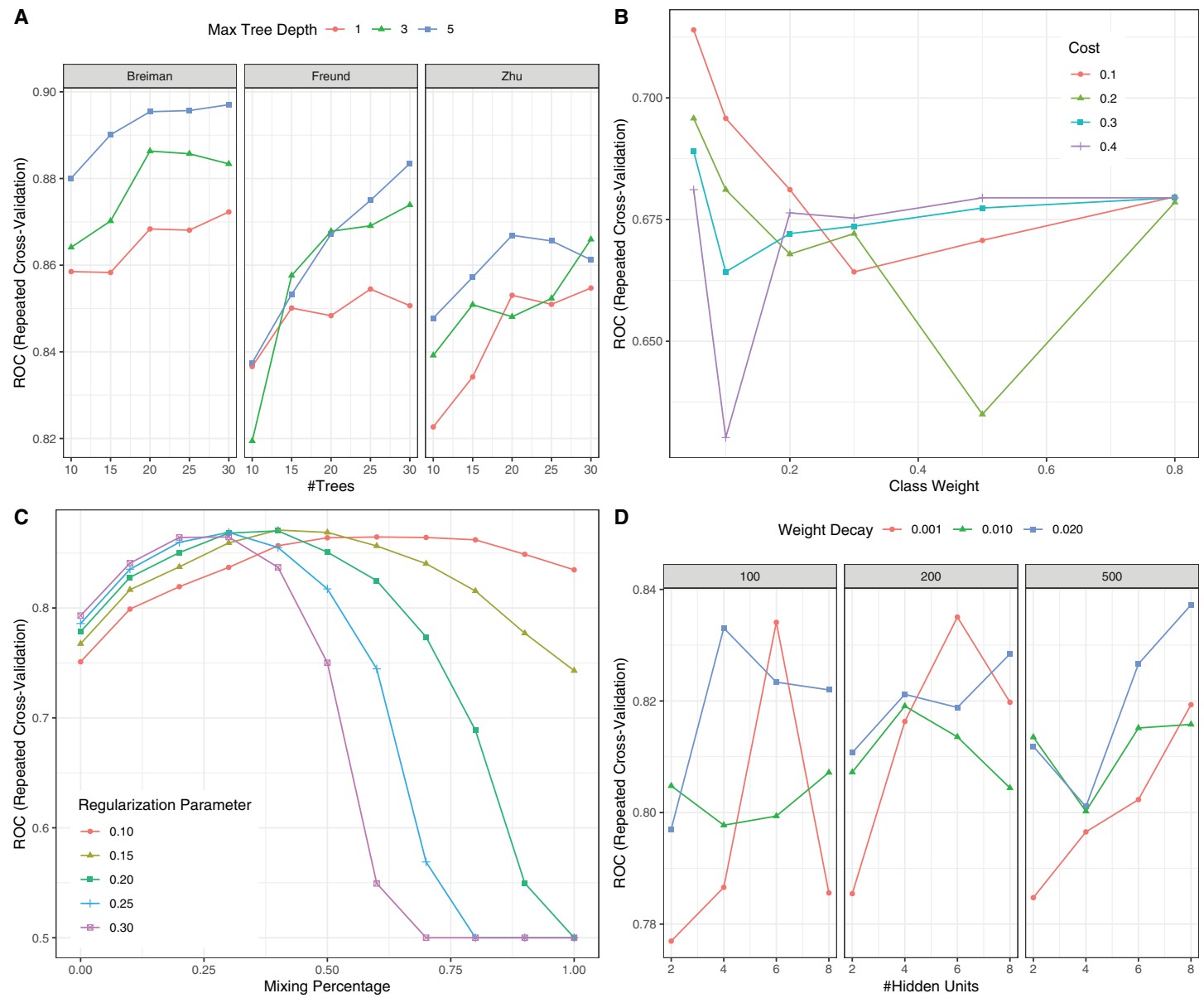

**Figure 6 Hyperparameter tuning for machine learning algorithms by grid search.** The area under the receiver operating characteristic curve (AU-ROC) was used as the performance metric, which was estimated by 5-fold cross validation method. (A) Hyperparameter tuning for Adaboost which included three hyperparameters: the number of iterations for which boosting is run or the number of trees to use, learning coefficient and maximum depth of the trees. In Breiman method, $\alpha = \frac{1}{2}\ln\left(\frac{1-error}{error}\right)$ is used. The Freund uses $\alpha = \ln\left(\frac{1-error}{error}\right)$. In both cases the AdaBoost.M1 algorithm is used and $\alpha$ is the weight updating coefficient. On the other hand, if the Zhu method is used, the SAMME algorithm is implemented with $\alpha = \ln\left(\frac{1-error}{error}\right) + \ln$ (number of class $-1$). (B) Hyperparameter tuning for SVM. The cost and class weight were tuned in a grid. (C) Hyperparameter tuning for generalized linear model with regularization. Hyperparameters included alpha (Mixing Percentage) and lambda (Regularization Parameter). (D) Hyperparameter tuning for neural networks. Abbreviations: Adaboost: adaptive boosting with classification trees; SVM: support vector machines.

## DISCUSSION

The study retrospectively analyzed a cohort of COVID-19 patients. The overall mortality rate was 17.1%. Risk factors for mortality risk included older age, decreased lymphocyte count, elevated LDH, troponin I and *D*-dimer. Two neural network models, NNet1 and

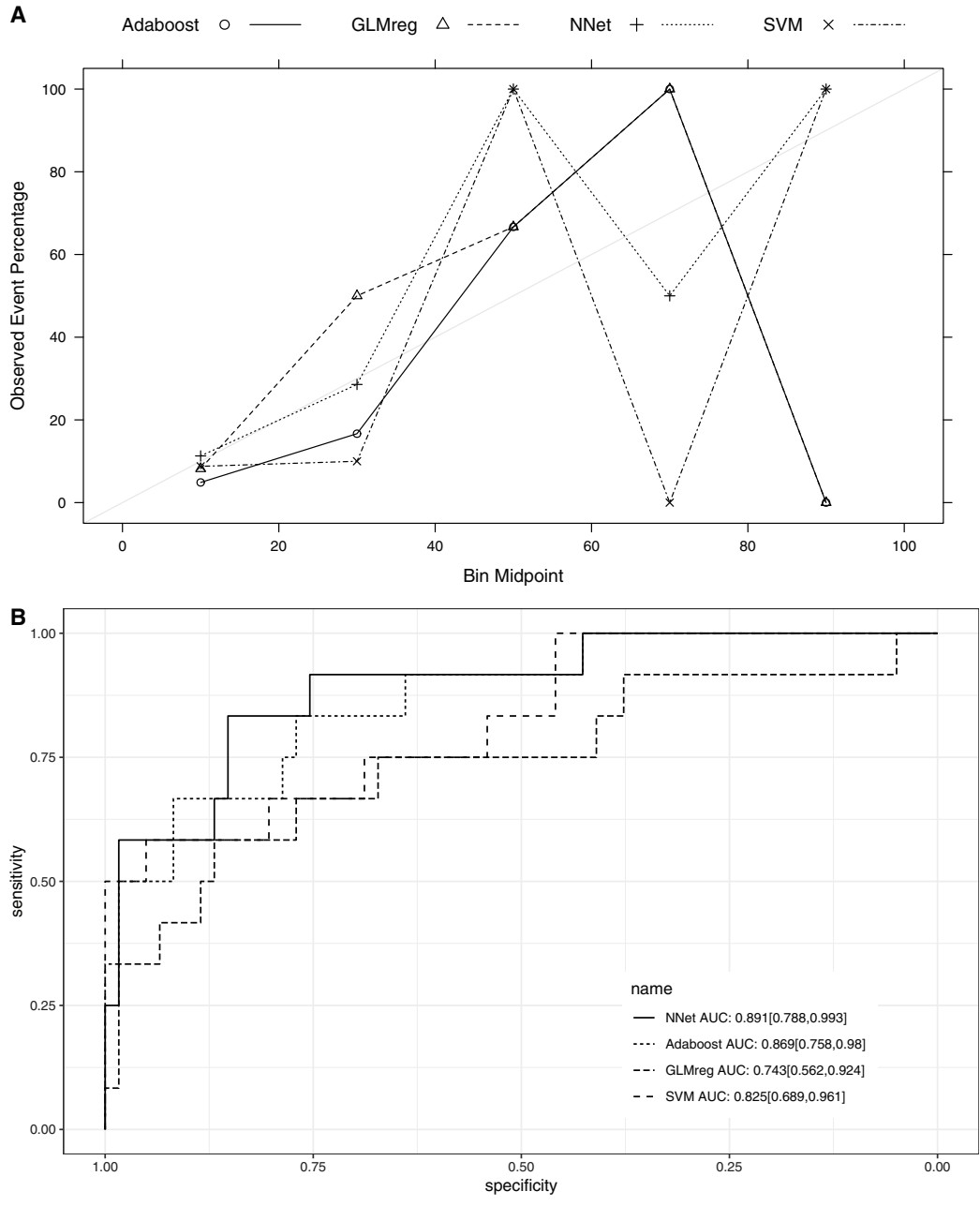

**Figure 7 Calibration (A) and Discrimination (B) of the models computed in the testing dataset.** Abbreviations: Adaboost, the adaptive boosting with classification trees; SVM, Support Vector Machines; NNet, neural networks; GLMreg, generalized linear model with regularization.

NNet2, which included nine and 32 predictors, respectively, were developed for risk stratification. Predictors were obtained on day 1 after hospital admission, which is useful for early risk stratification. The neural network models were found to be superior to the respective logistic regression models. These models were interpretable with the help of LIME.

The overall mortality rate of the study was higher than that in other studies; Guan WJ and colleagues reported that the mortality rate was 1.4% (*Guan et al., 2020*). However, their study included 552 hospitals across China. Other studies have shown that the mortality rate outside Wuhan is much lower than that in Wuhan (*Huang et al., 2020*; *Chen et al., 2020a*; *Wang et al., 2020*; *Xu et al., 2020*), probably due to the limited medical resources in Wuhan at the outbreak of the COVID-19 epidemic. The risk factors identified in our study cohort are generally consistent with other reports, including older age, more comorbidities, high SOFA score, and high *D*-dimer (*Zhou et al., 2020*). The strength of our study was its use of EHR, which allowed it to include many more variables (>100) than other studies. In conventional cohort studies, variables are included because domain knowledge determines whether they are related to clinical outcome. This approach is limited by human knowledge. In contrast, our study included all variables recorded in the EHR, which can help to identify unknown risk factors. One limitation of such a large-scale search of risk factors is the problem of multiple testing. We used a conservative *p* value of 0.001 to identify risk factors to reduce the false positive rate. Furthermore, a volcano plot was used to identify risk factors for mortality. The volcano plot identified risk factors in two dimensions with *p* value < 0.001 and SMD > 0. The former guaranteed that the difference was not due to chance, and the latter ensured a difference that was clinically relevant. We identified additional risk factors, such as *C*-reactive protein, albumin, albumin to globin ratio and neutrophil count.

The large number of variables also imposed great challenges such as multicollinearity, non-linear terms and high-order interactions for conventional statistical modeling. For example, the generalized linear model can capture interaction by explicitly designating the interaction term, but the number of variable combinations can be as high as $\times 10^9$/L, which is far from the computational power. Thus, we employed neural networks to automatically capture these high-order terms. The results showed that our neural network models NNet1 and NNet2 were better than the conventional logistic regression models. The employment of machine learning algorithms for the risk stratification of COVID-19 patients has never been reported upon. Given the potential pandemic outbreak of COVID-19 (*Eurosurveillance Editorial Team, 2020*), such models can be helpful for other countries.

Several limitations must be acknowledged in the study. First, the study was based on a single center, and the prediction model was not validated using an external cohort. Thus, further validation studies are required to see whether the model can accurately predict mortality outcome. The relatively small sample size can also explain the instability of risk factor rankings in NNet1 and NNet2. However, we have tried to prevent model overfitting by using the cross-validation method, in which model validation was performed in samples that were not used for training. Second, missing values existed in our EHR, which may introduce potential bias. Missing values per se may carry important predictive information. For example, our previous studies have shown that missing values on blood gas in hospitalized patients have better clinical outcomes (*Zhang et al., 2019*). In the study, we assumed that the missing values were produced at random, and imputation with random sampling was performed. Third, while the use of first-day biomarkers allows

for early prediction of mortality outcome, it is at the cost of model accuracy. The clinical trajectories can be quite different for patients with similar conditions on first admission, as a variety of interventions/procedures may lead to different outcomes. In such a situation, the performance of the model including variables on day 1 can be compromised. Finally, the mortality used in the study is all-cause mortality, we could not fully determine the specific causes of death in retrospective dataset.

## CONCLUSIONS

In conclusion, our study included a cohort of COVID-19 patients with a mortality rate of 17.1%. Several risk factors were identified considering both clinical and statistical significance. Some novel risk factors such as BNP, Hypersensitive troponin I and Creatine kinase isoenzyme. We further developed two neural network models, with the variables selected by using GA. The model performs much better than the conventional generalized linear models. Due to the worldwide outbreak of the COVID-19 pandemic, the study can provide risk stratification tool for the triage and management of these patients. The medical resources can be allocated to the most critically ill patients.

## ABBREVIATIONS

| | |
|---|---|
| COVID-19 | Coronavirus Disease 2019 |
| ICU | Intensive care unit |
| GA | Genetic algorithms |
| IQR | interquartile range |
| NNet1 | Neural network model 1 |
| NNet2 | Neural network model 2 |
| AUROC | Area under the receiver operating characteristics |
| CI | Confidence interval |
| EHR | electronic healthcare recordss |

### Funding

Hongying Ni received funding from the Jinhua Novel Coronavirus Pneumonia Emergency Response Research Project (2020XG-03). Zhongheng Zhang received National Natural Science Foundation of China (Grant No. 81901929). The funders had no role in study design, data collection and analysis, decision to publish, or preparation of the manuscript.

### Grant Disclosures

The following grant information was disclosed by the authors:
Jinhua Novel Coronavirus Pneumonia Emergency Response Research Project: 2020XG-03.
National Natural Science Foundation of China: 81901929.

### Competing Interests

The authors declare that they have no competing interests.

## Author Contributions

- Yuetian Yu conceived and designed the experiments, prepared figures and/or tables, and approved the final draft.
- Cheng Zhu conceived and designed the experiments, performed the experiments, authored or reviewed drafts of the paper, and approved the final draft.
- Luyu Yang analyzed the data, prepared figures and/or tables, and approved the final draft.
- Hui Dong performed the experiments, analyzed the data, authored or reviewed drafts of the paper, and approved the final draft.
- Ruilan Wang performed the experiments, authored or reviewed drafts of the paper, and approved the final draft.
- Hongying Ni performed the experiments, authored or reviewed drafts of the paper, and approved the final draft.
- Erzhen Chen performed the experiments, analyzed the data, prepared figures and/or tables, and approved the final draft.
- Zhongheng Zhang conceived and designed the experiments, analyzed the data, prepared figures and/or tables, and approved the final draft.

## Human Ethics

The following information was supplied relating to ethical approvals (i.e., approving body and any reference numbers):

The study was approved by the ethics committee of Wuhan Third Hospital (KY2020-007) and was conducted in accordance with the Helsinki Declaration.

## Ethics

The following information was supplied relating to ethical approvals (i.e., approving body and any reference numbers):

The study was approved by the ethics committee of Wuhan Third Hospital (KY2020-007) and was conducted in accordance with the Helsinki Declaration.

## Data Availability

The raw data are available in the Supplemental Files.

## Supplemental Information

Supplemental information for this article can be found online at http://dx.doi.org/10.7717/peerj.9885#supplemental-information.

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
