# Peer review of "Identification of risk factors for mortality associated with COVID-19"

_PeerJ, doi:10.7717/peerj.9885_

## Round 0.1 · original submission · Major Revisions

Thank you for submitting the manuscript titled "Early prediction of mortality in COVID-19 patients: Development of neural network models by genetic algorithms" for possible publication in PeerJ. Your manuscript has been handled by me (Tuan Nguyen), and has been reviewed by two experts in the field. The experts' comments are attached for your perusal. As you will see, both reviewers recognize the utility of your study, but they also raise a number of issues concerning the interpretation and data analysis. I invite you to address their comments.

I have also read your manuscript, and thought that it may be publishable if you can please address the following points:

1. I am not convinced that your paper is qualified as a prediction model. First, your study was basically a retrospective study, not prospective study which is usually required for a prediction model. Second, your models have not been validated (internally or externally) which is commonly required for a prediction model. Third, your models lack transparency (more on this later). Fourth, the sample size was not adequate for a proper development of prediction model. Therefore, I strongly suggest that you change the title and the focus of the paper to an association study. Perhaps the best you could claim is factors that are associated with mortality. Your paper title may read (just a suggestion) "Identification of risk factors for mortality associated with COVID-19".

2. I am concerned about the ascertainment of mortality. How was mortality ascertained? Was it based on 30-day mortality? How did you ascertain that a death was causally related to COVID-19? I am a bit concerned that D-dimer was related to mortality, suggesting that neurological disorders might have contributed to mortality. This is an important issue, and I strongly suggest that you provide much more detailed description.

3. Could you please explain how did you arrive at the sample size of 246. This question was also raised by a reviewer. You have 32 predictor variables, but the number of events (eg mortality) was 42, which raises the problem of over-fitting and lack of statistical power.

4. The transparency of modelling is a big problem of this manuscript. In theory, NNet model can take into account the non-linear and interactions between predictor variables, but I (and readers) cannot see those interactions. Actually, even the logistic model reported in the paper is also not clear to me. I would like to see full estimates of parameters in the model and their statistical significance.

5. I am concerned that normally distributed data were described by median and interquartile range, but non-normally distributed data were described by mean and standard deviation. This is wrong. Could you please provide a table (perhaps in the appendix) showing full descriptive statistics (eg mean, SD, median, IQR) for ALL variables and stratified by mortality status.

6. In Fig 4, you stated that "NNet1 is the neural network model including 9 predictors, and it outperforms the logistic regression models (Logit1 and Logit2)." However, the AUC for NNet1 was 0.806, almost identical to Logit2 model (AUC 0.802). I suggest that you conduct a significance test to test for the difference in AUC between models.

Reviewer 1 ·

Basic reporting

The paper is generally well written, but needs to avoid repetition (e.g., The sentence, “The mortality rate was 17.1%”, repeated several times). It provided enough background, literature reviewing, and references. The manuscript is well organized and easy to follow. Figures and tables are all in the right format. In the manuscript, it is claimed: “the datasets used and/or analyzed during the current study are available from the corresponding author on reasonable request”. However, the paper does not have a significant contribution in theory; the major contribution of this paper is on the application side, which is heavily relying on the data. I am not quite sure if the availability of data is conditional, will it influence the acceptance of the paper?

Experimental design

This is an original primary research within the scope of the journal. The research question is well defined and meaningful. The statistic results and the established predictors may provide valuable tools for the identification of vital status at the hospital discharge about COVID19. The technical and ethical standards of the paper are high. It also provided enough details about experimental setting and data collection.

Validity of the findings

The benefit of the research is significant as it directly addresses some critical issues of the COVID19. The statistical analysis is good. The conclusion is too brief; it is just a simple repetition of the introduction. Therefore, the conclusion should be rewritten.

Additional comments

This paper provided statistical analysis and established predictors for COVID 19, which may provide valuable tools for the identification of vital status at the hospital discharge. However, this paper has not provided sufficient details of the major outcomes. For example, there is no description of NNet1 and NNet2. If they are general neural network classifiers constructed by the authors, the author should specify the type of them (e.g. RNN or CNN), and provide the details of its hyperparameters, e.g., number of layers. If the authors adopted the NNet from somewhere, the authors should provide where it comes from. Also, as the dataset is not big, the authors should justify the reason to use a neural network as the predictor. Instead of comparing them with a linear classifier, the author should compare the performance of the neural network classifiers with the popular nonlinear classifiers, e.g., SVM. The conclusion is too brief; it is just a simple repetition of the introduction. Therefore, the conclusion should be rewritten.

Reviewer 2 ·

Basic reporting

The biggest issue is the use of a relatively small patient set (N=246) from a cohort with unusually high mortality rate (>17%) and the lack of an independent patient set for validation.

A strength of the study is that it uses a large number of variables (~100) to assess mortality using a machine learning approach. However, there is no clear conclusion whether or not new variables associated with mortality have been identified.

Experimental design

The biggest issue is the use of a relatively small patient set (N=246) from a cohort with unusually high mortality rate (>17%) and the lack of an independent patient set for validation.

Machine learning models were then trained on what I assume is data from 20% of the patients (i.e. ~50) while the prediction accuracy was evaluated using data from the remaining 80% (ie. ~197). I wonder if this represents a balanced analysis given that only 17% of all patients (i.e. ~8 in the training and ~33 in the testing set) did succumb to Covid-19 ?

Validity of the findings

The authors show that their neural nets algorithms performed better compared to their linear models when predicting mortality to Covid-19. Interestingly, their NNet1 and NNet2 algorithms partly assessed unique variables e.g. ‘α.hydroxybutyrate.dehydrogenase’ was the second most informative feature in NNet2 (total of 32 variables) while it was not used at all in NNet1 (total of 9 variables). It is thus unclear to me what this means for the utility of ‘α.hydroxybutyrate.dehydrogenas’ in predicting risk ? On the contrary, are variables the feature in both models (i.e. lactate.dehydrogenase) considered to be most predictive for mortality to Covid-19 ?

I would thus encourage to authors to extend their relatively short section which aimed to interpret the results from the NNet1 and NNet2 algorithms.

Most critically, it remains unclear how this model performs on other independent datasets and especially those with lower mortality rates ?

---

## Round 0.2 · Minor Revisions

Thank you for addressing the reviewers' and my concerns. Your revised manuscript has been re-reviewed by a reviewer and myself, and as you will see from the reviewer's comments, there is a minor issue that I hope you can address.

Reviewer 1 ·

Basic reporting

Although there are still some typographical errors, the revised document and the response letter seem better. The literature review was good, and the original data was shared this time. A comparative study was carried out using other nonlinear classifiers. My concern is that the NNet1 and NNet2 proposed are not deep neural networks, but their performance is better than most other nonlinear classifiers. The author must provide a convincing explanation in the next round of review. Actually, if the parameters of other classifiers are not adjusted properly, the comparison results may be biased.

Experimental design

The revised version for experimental design is acceptable with methods described in sufficient detail. The revised title more properly reflects the targets of the paper.

Validity of the findings

The revised version addressed most of my concerns. The only issue is the novelty of the paper. Although the data is precious, in terms of classification methodology (normal neural network classifier with existing parameter tuning methods), no significant contribution can be detected. I am not sure whether this is acceptable in PeerJ.

Additional comments

The revised document and its response letter resolved most of my concerns. A comparative study was conducted and the results of other non-linear classifiers were provided. My concern is that the Nnet1 and NNet2 proposals are not deep neural networks, and their performance is better than most other nonlinear classifiers. The author must provide a convincing explanation in the next round of review. In addition, if the parameters of other classifiers are not adjusted properly, the comparison results may be biased.

---

## Round 0.3 · accepted · Accept

Thank you for addressing the reviewer's and my comments. I am happy that the revised manuscript is now acceptable for publication in PeerJ.